# In Vitro Hydroxyapatite Nucleation in Cationically Cured Epoxy Composites with Pulverized Date Seed

**DOI:** 10.3390/polym16243463

**Published:** 2024-12-11

**Authors:** Muhammad Atif, Hafsah Akhtar, Muhammad Imran, Muhammad Zafar Ullah, Hina Andaleeb, Muhammad Asif Hussain

**Affiliations:** 1Chemistry Department, University of Education Lahore, Vehari Campus, Vehari 61100, Pakistan; 2Interdisciplinary Research Centre in Biomedical Materials (IRCBM), COMSATS University Islamabad (CUI), Lahore Campus, Lahore 54000, Pakistan; 3Research Center for Advanced Material Science, Chemistry Department, Faculty of Science, King Khalid University, P.O. Box 9004, Abha 61413, Saudi Arabia; 4Department of Biochemistry, Bahauddin Zakariya University, Multan 60800, Pakistan; 5Institute of Metallurgy and Materials Engineering, University of the Punjab, Lahore 54000, Pakistan

**Keywords:** cationic UV curing, biomaterials, date seed, nanocomposites, hydroxyapatite, epoxy

## Abstract

Recently, driven by a growing focus on environmental sustainability and cost-effectiveness, researchers have shown a keen interest in creating useful materials from bio-wastes, particularly for their potential applications in the biomedical field. Current research has been conducted on the impact of date seed powder (DSP) on hydroxyapatite (HA) formation, specifically in relation to the promotion of bone health and regeneration. HA is an essential component of bone tissue and plays a crucial role in maintaining bone strength and structure. Date seed (DS) was used in two forms i.e., grains and powder, with unmodified and modified surface chemistries. Prepared composites were tested in vitro by soaking them in simulated body fluid (SBF). X-ray Diffraction (XRD) and Fourier Transform Infra-Red (FTIR) confirmed HA formation in all soaked samples. Thermogravimetric analysis (TGA) results indicated an improvement in thermal stability after soaking, suggesting a higher concentration of HA. Unsoaked samples were observed to have higher heat flow than soaked samples. The high gel content (GCs) over 90% and low hydrophilicity (less than 5%) of DSP-based composites were proven to be beneficial in HA nucleation. Antibacterial activity showed that the addition of DS filler yielded superior results compared to the pristine sample. Additionally, the modified samples demonstrated better antibacterial results than the unmodified ones.

## 1. Introduction

In recent years, bone deterioration has increased. Annually, 250,000 wrist fractures, 7,000,000 vertebral fractures, and 2.8,000,000 hip fractures were found to have occurred in the US, costing USD 10 billion [1]. After femur fracture therapy, 25–42% of patients had osteoporosis and infection [2,3]. Bone regeneration, restoration, and replacement techniques have emerged as a critical area in orthopedics [4]. The development of orthopedic implants has evolved over time, starting with metals [5,6], transitioning to alloys [7,8], and eventually to polymer composites, i.e., PEEK and epoxy [9] with different fillers [10,11].

Epoxy resin is extensively utilized as the structural foundation of composite materials, owing to its exceptional mechanical characteristics and bonding capabilities. Epoxy polymerization through thermal curing is a conventional method, whereas cationic polymerization stands out as a distinctive process, utilized for its effectiveness in generating highly cross-linked polymer networks [12]. These networks provide necessary mechanical strength, chemical stability, and durability required for biomedical applications. Choosing cationic polymerization over other methods, such as free-radical polymerization, is driven by its ability to proceed at lower temperatures and its compatibility with a wide range of fillers and additives. Cationically polymerized integration of surface modified pulverized DS into the epoxy matrix is a novel approach to enhance HA nucleation within the composite. This polymerization process thoroughly disperses surface-modified DS particles inside the epoxy matrix, which leads to their full integration as a constituent phase of the finished composite material. This dispersion is crucial for ensuring that the natural fibers and particles from the DSP and date seed grains (DSG) are available as nucleation sites for HA formation. The polymerization process has been finely tuned to achieve a homogeneous distribution of these particles, which directly impacts the performance of composites. A comprehensive review of the existing literature on epoxy-DS composites revealed a significant research gap (Table 1): no studies investigated the combined effects of DSP modification and photochemical curing. This lack of exploration presented an opportunity for innovative research in this line.

Currently, bioactive composite bone implants with surface modifications are being used in clinical practice [17]. According to existing research [18,19], DSP has been found to include a range of chemicals, including calcium, phosphorus, and different bioactive molecules, which have the potential to facilitate HA nucleation. These compounds are crucial for the provision of vital mineral elements and organic ingredients that are indispensable for HA crystal formation and development. Changes in filler composition leads to noticeable variations in properties, as evidenced by chemical and experimental analysis [20]. This research had an initial emphasis on evaluating the suggested role of DS material as a filler in the development of polymer composites for HA crystal nucleation, as well as on investigating their impact on improving antibacterial activity. Furthermore, the examination of the surface modification of biobased material (DSP and DSG) is essential in understanding its impact on the interactions among components of composites [20]. According to Bagheri, [21] epoxy composites must be thermally cured under certain pressure conditions, but in this study, composites were photo-cured through unique UV curing, an eco-friendly approach [22]. This method of composite preparation increased curing conversion and decreased the curing time of composites [22,23].

## 2. Experimental

### 2.1. Materials

The DS was purchased locally. The BPADGE monomer (Figure 1a), triarylsulfoniumhexafluorophosphate (50% soln in propylenecarbonate), photo-initiator (PI Figure 1a), and H_2_O_2_ (35%) modifier were purchased from Sigma Aldrich, Darmstadt, Germany). The NaCl (AnalaR NORMAPUR, Lutterworth, UK), NaHCO_3_ (AnalaR NORMAPUR, Lutterworth, UK), KCl (Riedel-deHaen, Seelze, Germany), K_2_HPO_4_·3H_2_O (Sigma Aldrich, Darmstadt, Germany), MgCl_2_·6H_2_O (Sigma Aldrich, Darmstadt, Germany), CaCl_2_ (AnalaR NORMAPUR, Lutterworth, UK) and Na_2_SO_4_ (AnalaR NORMAPUR, Lutterworth, UK) were used in SBF solution. Tris-hydroxymethyl aminomethane and 1M HCl were used to maintain SBF pH.

### 2.2. Methods

#### 2.2.1. Filler Preparation

The DSs were washed with deionized water and dried under sunlight for 2 days. The DSP and DSG were prepared by mechanical crushing and grading of size <125 and 300–400 microns, respectively (Figure 1b).

#### 2.2.2. Filler Modification

The DSP/DSG were modified with 20 wt.% H_2_O_2_ in water bath at 35 °C for 24 h, and oven dried at 60 °C for 24 h. The modified samples were named as DSMP for modified powder, and DSMG for modified grains (Figure 1b).

#### 2.2.3. Composite Preparation

The composites were photo-cured through an eco-friendly approach [22], as per composition in Table 2. PI was irradiated under room temperature in air for 30 s with a UV lamp (ML-3500S, MAXIMA ™ 3500 Series, Spectro-UV, New York, NJ, USA) having 50,000 uW/cm^2^ intensity of 365 nm. Irradiated PI was mixed with monomer-filler (5 wt.%) mixture (Figure 1c). The curing time was observed through tacking behavior. No solvent, including water, was utilized during sample preparation through the photo-curing process. Unless there was a very slight interaction between the sample material and the air moisture, the samples were fully dry after achieving tack-free behavior. The samples were named as PC for photocured pristine epoxy, PGC for photocured grain composite, PMGC for photocured modified grains composite, PPC for photocured powder composite, and PMPC for photocured modified powder composite.

### 2.3. SBF Immersion Test

The prepared samples (Figure 1c) were soaked in SBF for four weeks at 36.5 °C. The samples were labeled 1, 2, 3, and 4 according to the number of weeks they were soaked. SBF with composition equal to blood plasma was prepared according to the procedure developed by Kokubo et al. [24]. The samples were tested weekly for HA crystal formation, after thorough rinsing with distilled water and drying at 60 °C.

### 2.4. Characterization of Prepared Samples 

The fillers and composites were characterized separately. The filler surface chemistry was examined by FTIR-Attenuated Total Reflectance (FTIR-ATR, IR spirit, Shimadzu, Japan). pH and conductivity were probed at room temperature by dispersing 0.005 g in 10 mL distilled water, stirred for 3 h, and pH/conductivity was determined with a Milwaukee combo meter (MW805 MAX 4-in-1 pH/EC/TDS/Temp, Milwaukee Instruments, Rocky Mount, NC, USA). The acid contents (*n_CSF_*) of particles were analyzed through direct Boehm’s titration (Equation (1)) [20].
(1)nCSF=B×VB−HClVHCl×VBVa

FTIR-ATR (IR spirit, Shimadzu, Japan) was also used to analyze HA nucleation in composites before and after SBF soaking for 7, 14, 21, and 28 days. TGA-DSC tests were performed with a Simultaneous Thermal Analyzer (STA, SKZ-1060, SKZ Ltd., Jinan, China) for the determination of the thermal characteristics of materials at a temperature range of 25–500 °C with a heating rate of 10 °C/min, in Al crucibles, using air as purge gas at a flow rate of 10 mL/min. For the TGA/DSC test, samples were taken in the range of 8–11 mg. A microscope (IM-910, IRMECO, Lütjensee, Germany) with a WF 10×/20 mm wildlife eyepiece was used to capture images. Surface morphologies of the samples were performed using a variable pressure Field Emission Scanning Electron Microscope (FE-SEM, Apreo-S, Thermo-Fischer, Eindhoven, The Netherlands). GC was measured for all samples by weight loss in 10 mL DCM, and was calculated by using Equations (2) and (3) [22].
(2)$$Extract\;\%=\frac{W_{s}-W_{d}}{W_{s}}\times 100$$
(3)GC %=100−Extract %
where *W_s_* is the initial weight of sample and *W_d_* is the difference in sample weight before and after GC.

The water adsorption capacity (*WAC*) of the samples was measured by keeping them in deionized water (pH 7.1) at room temperature for a week. After one week, the samples were removed from the water, wiped with a soaking cloth and then oven dried at 60 °C for 15 min [25], and weighed to calculate the *WAC* percentage by using Equation (4).
(4)$$WAC\;\%=\frac{W_{t}-W_{o}}{W_{o}}\times 100$$
where *W_t_* is weight of composite after water adsorption and *W_o_* is weight of original composite.

The crystalline and amorphous behavior of pristine material, unsoaked, and soaked composites was determined by using X-ray diffraction (XRD, Bruker AXS D8, Kontich, Belgium) at a 40 kV voltage and a 30 mA current with Cu Kα radiation (1.540 Å), in the 2θ range of 5–115°.

The sample flow was determined by placing one drop of properly mixed sample on a glass slide and measuring its diameter (Table 2). Then a second slide (5 ± 2 g) with a load of 115 g was placed on it for 7 min. After the removal of the applied load minor and major, the diameter was measured with a digital Vernier Caliper (150 mm TMT331501, Total, Putian, China). The test for flow was repeated if the mixture did not spread in a circular way [26]. By gingerly contacting with a finger till tacking was observed, tack free time was noted [27]. The antibacterial activity of PC, PPC, PGC, PMGC, and PMPC (2.5 g/100 mL) was tested against a gram-negative bacterium, *E. coli*, using the broth dilution method [28]. A fresh culture of *E coli* was prepared using Luria-Bertani (LB) broth media and the samples were incubated with *E. coli* at 37 °C. After overnight incubation, the relative percentage inhibition (% *Inhib*) of bacterial growth was calculated by measuring an optical density at 600 nm. Gentamicin was used as the positive control while the negative control was left without any sample or any antibacterial compound added to it.
(5)$$\%\;Inhib=(Control\;OD-(\frac{Sample\;OD}{Control\;OD}))\times 100$$

The antioxidant activity of samples was assessed using the DPPH (2,2-diphenyl-1-picrylhydrazyl) free radical scavenging method. This assay demonstrates the materials’ capacity to donate hydrogen atoms or electrons to neutralize DPPH, a stable free radical that causes a purple to yellow color shift. To each sample (5 mg/mL) was added 0.1 mM solution of DPPH in methanol. The mixture was incubated in the dark at room temperature for 30 min to allow the reaction to occur. After 30 min of incubation, the absorbance was measured. The percentage for DPPH radical scavenging activity was calculated by using Equation (6).
(6)$$DPPH\;radical\;scavenging\;activity=\frac{Absorbance\;of\;Control-Absorbance\;of\;Sample}{Absorbance\;of\;Control}\times 100$$

## 3. Results and Discussion

Epoxy composites with DS fillers were prepared using the SMART photo-curing approach. A thorough analysis of the fillers’ surface chemistry effect on composite properties was made based on the following findings.

### 3.1. Filler Analysis

FTIR of unmodified and modified DS grains and powder (Figure 2a) showed characteristics peaks at 3410 cm^−1^ indicating OH stretching of alcoholic compounds and carboxylic acids, at 1675 cm^−1^ for the carbonyl group, and at 2800 cm^−1^ to 3000 cm^−1^ for the CH stretching of methyl and methylene groups [29,30]. The smaller particle size of the powdered samples, and hence the higher concentration of functionalities near the surface, may explain why the peaks were prominent in these samples.

#### Physico-Chemical Analysis

A negative correlation between pH and conductivity in all samples was observed. In comparison to their granular counterparts, the powdered samples had a higher pH and lower conductivity. This might be attributed to high surface functionalities as shown by FTIR.

### 3.2. Composite Analysis

The intensification of FTIR peaks were observed with high soak time in all five samples (Figure 2a). SBF-treated samples showed an increase in the PO_3_^−4^ peak (1031 cm^−1^), which was absent in the unsoaked counterpart. At 1509 cm^−1^, 2922 cm^−1^, and 3400 cm^−1^, respectively, CO_2_^−3^, CH, and HO stretching was observed [31]. Saturation times of 7, 14, 21, and 28 days resulted in increasingly jagged peaks.

#### 3.2.1. Morphology

Different levels of HA crystallization were observed in photo-cured composites subjected to SBF treatment versus those that were not (Figure 3); for example, in unsoaked samples there was no apatite formation, while in SBF-treated samples there was a very clear network of HA crystallization after 4 weeks. Apatite was increased in composites subjected to extended SBF soaking times, suggesting that this mineral is a byproduct of the SBF soaking process.

The generation of HA crystals was observed more successfully in modified powder samples (PMPC) compared to the control counterparts made with unmodified powder (PPC). On the contrary, granular samples of both modified and unmodified compositions (PGC and PMGC) showed observable crystal formation.

The morphology of composites was examined by microscopic analyses (Figure 3). Microscopic images of unsoaked samples were analyzed alongside those of samples soaked for 1, 2, 3, and 4 weeks, revealing a distinct enhancement in HA nucleation corresponding to prolonged soaking duration. The images provided convincing evidence of HA crystal growth on SBF-soaked composites. Cross-sectional SEM revealed textured surfaces of composites prior to immersion, whereas the immersed composites exhibited relatively smooth surfaces along with indications of HA nucleation, as observed in PC and PMGC-4. Based on these observations, it can be inferred that HA in the soaked samples was thoroughly integrated within the matrix. Similarly, surface SEM revealed an enhancement in surface roughness, thereby corroborating HA nucleation. Microscopic images and SEM analysis revealed a notable tendency for HA crystallization in all SBF soaked samples, encompassing both the pristine epoxy and DS-epoxy composites, in contrast to the unsoaked samples. This illustrates the ability of the epoxy matrix to initiate the nucleation of HA. Composites that incorporate grains (PGC and PMGC) demonstrated a reduced crystal count in comparison to powder composites (PPC and PMPC), likely attributable to a more limited available space for the onset of HA nucleation. The incorporation of DS as a filler positively influenced certain attributes of composites, including water absorption capacity, thermal dissipation, and antibacterial properties.

#### 3.2.2. Thermal Analysis

The comparative thermal stability of prepared composites was analyzed with TGA (Figure 2b).

The addition of filler increased heat capacity or decreased thermal dissipation in the polymeric network, both of which contributed to improved thermal stability. Better thermal stability of polymeric networks was seen when fillers were in powdered form. This may be because the filler disperses well in the polymer matrix and effectively regulates thermal dissipation.In contrast to unmodified powder composites, high thermal degradation in modified powder composites was observed below 400 °C, which may be a result of the increased surface oxidizable species present in modified fillers. However, above 400 °C, the degradation rate reduced in modified powder composites, which may be due to HA crystal formation.It was found that the thermal stability of unsoaked composites is lower than that of SBF-soaked samples, which is consistent with the creation of a HA network in the latter.

DSC analysis (Figure 2c) demonstrates distinct heat flow profiles for unsoaked and soaked samples, suggesting modifications in their structural properties. Unsoaked samples exhibit elevated heat flow values, suggesting a structural chemistry that enables greater heat dissipation through the materials, whereas 28 days-soaked samples were observed to have the least heat flow, which might be attributed to higher HA contents. Unsoaked samples were observed for T_m_, indicating the semi-crystalline nature of the sample, whereas soaked composites showed either an absence (grain composites) or reduction (powder composites) in the peak intensity. This exothermic peak represents molecular chain rearrangement in amorphous domains [32]. The peak’s absence or diminution due to the absence of molecular chain rearrangement in the amorphous domain suggests the formation of HA crystals.

#### 3.2.3. Physico-Chemical Analysis

To find out curing and hydrophilic behavior of the materials, their GCs and WACs were observed (Table 2). After being soaked, pristine epoxy adsorbed more water and had lower GCs, both of which are indicative of less crosslinking and more hydrophilic interaction in the sample.

In unmodified granular composites, both GCs and WAC increased after soaking, signposting augmented crosslinking as well as enhanced hydrophilic interaction of the material, whereas, modified granular samples revealed an increase in GCs but a decrease in WAC, indicating enhanced crosslinking but less interaction with water. These results demonstrate that in modified granular composites, HA crystals developed concordantly inside the polymeric network, protecting the integrity of the crosslinking molecules.

When it comes to powder filler composites, both the modified and unmodified samples showed an increase in GC and a decrease in WAC, indicating that the polymeric network formed well and the hydrophilic response reduced. This could be because of the filler’s high surface chemistry and uniform distribution inside the polymer matrix.

In this study, BPA DGE was taken as monomer and 5 wt.% DS powder/grains (modified and unmodified) were added as fillers. Flow values of the pure monomer were observed to be 23.44 mm (Table 2); however, all experimental groups demonstrated 4–6 folds drop in flow value as compared to the pure monomer. Samples with unmodified granular fillers had a lower flow value (4.21 mm) than their modified counterparts (5.02 mm), which may be attributable to the smaller size of the modified samples (Figure 1d). Conversely, the flow value of the sample with modified powder filler was lower (5.03 mm) than that of the unmodified counterpart (6.85 mm). This discrepancy may stem from an enhancement in the surface chemistry of the filler, which ultimately leads to a more robust interaction between the filler and the matrix. In examining the disparity in flow values between composites incorporating powders (high flow) and grains (low flow), one might consider particle size as a pivotal factor, which is inversely related to surface area and, consequently, to the interaction between the filler and the matrix.

During the curing process, the addition of filler reduced tack-free time of the composites compared to the pure monomer. This fast curing kinetics is evidence that filler reacted with the monomer to speed up the curing process. Composites with granular filler have a longer tack-free time than those with powder filler. The shortest curing time was observed in composites with modified powder filler (Table 2).

#### 3.2.4. Structural Parameters

XRD analysis of the prepared pristine and composite samples before and after 4 weeks of soaking in SBF is presented in Figure 2d and Table 3. Peaks indicated a crystalline pattern within the polymeric matrix even prior to soaking, showcasing a consistent arrangement within photo-cured epoxy composites. A comparison of samples without filler (PC) and composites with unmodified fillers (PGC and PPC) indicates enhanced crystallinity, likely due to the structural characteristics of the added fillers. A comparative analysis of composites utilizing unmodified fillers against those employing modified fillers demonstrated a significant enhancement in peak intensity, suggesting that modification favorably influenced surface chemistry, hence reinforcing the crystalline structure in polymeric composites.

After soaking in SBF solution for 28 days, XRD patterns depicted HA crystals formation. The peaks’ intensity of HA on all samples after 28 days was significantly stronger than that on unsoaked samples, indicating that HA was formed after 4 weeks of soaking. The results showed that the corresponding peaks were assigned to (110), (111), (211), and (104) planes of HA [26,32]. In the XRD pattern, the width of the peaks is inversely proportional to the crystal size. A thinner peak corresponds to a bigger crystal. A broader peak at the start of the spectra means that the sample is amorphous in nature, a solid lacking perfect crystallinity. This was consistent with the results obtained from the FTIR spectra and SEM images.

#### 3.2.5. Antibacterial Activity

The percent inhibition against *E. coli* bacteria revealed significant differences between the pristine sample (PC) and its composites with DS fillers. The lack of DS filler facilitated significant bacterial growth noted in the PC, which contained no filler at all. In contrast, all other samples containing DS fillers demonstrated superior bacterial growth inhibition compared to the PC. Pulverized DS makes these composites more resistant to bacteria due to the presence of ethyl acetate and other phenolic compounds like gallic acid, ferulic acid, p-coumaric acid, vanillic acid, and syringic acid [33].

The results for the initial set of samples (PC, PPC, PGC, PMGC, and PMPC) were compared with four-weeks SBF-treated samples (PC4, PPC4, PGC4, and PMGC4), as shown in Figure 4a and Table 4. In the initial set, PMGC exhibited the highest inhibitory activity (56.64%) indicating a suitable surface chemistry regarding antibacterial activity. This performance slightly reduced in the PMGC-4 variant, indicating potential temporal or stability-related effects on its antibacterial activity. PPC and PMPC showed moderate inhibition, while PC and PGC showed the least inhibition level. The data revealed that, modified filler composites presented better antibacterial behavior than unmodified filler composites because of the presence of ethyl acetate in pulverized DS. A reaction of ethyl acetate with H_2_O_2_, may give rise to peroxy acids that could be a major cause of inactivation of gram (-ve) bacteria [34]. The activity of PMGC across batches make it a promising candidate for further development in antimicrobial applications, particularly in tissue or bone engineering.

#### 3.2.6. Antioxidant Activity by DPPH Assay

The comparison of DPPH scavenging percentages (Figure 4b and Table 4) revealed that PC and composite samples demonstrated differing levels of antioxidant activity, contingent upon their composition. Among all tested samples, PMPC-4 demonstrated superior results, exhibiting enhanced antioxidant activity relative to its counterpart, PMPC. The composite containing modified DSP exhibited superior DPPH scavenging results compared to the composites with DS grains. This improvement was attributed to the favorable surface chemistry of DSP, which enhances the surface area available for interaction with the media. PMPC may serve as an effective agent for scavenging free radicals.

#### 3.2.7. HA Formation: Effect of DS and Epoxy Polymer

Observations indicate that inclusion of DS as a filler has a beneficial effect on HA formation, potentially due to acetate ions in pulverized DS. Acetate ions function as binding agents for Ca ions found in SBF. Epoxy polymer (glycidyl ether) facilitates the incorporation of Ca ions into the interstices, thereby enabling an appropriate interaction with acetate ions. Subsequently, hydroxyl and phosphate ions encircle the metal ions and ultimately precipitate as HA (Figure 1, Equation (7)).
(7)$${10Ca}^{2+}+{6PO}_{4}^{3-}+{2OH}^{-}\to {Ca}_{10}{(PO}_{4}{)}_{6}{\left(OH\right)}_{2}$$

## 4. Conclusions

In this research, a convenient and simple approach was successfully implanted for surface modification of filler particles prepared from DS. Surface modification produced a range of surface chemistry. After surface modification, strong filler-matrix interaction was observed through sample flow, which was five folds less than that of a standard monomer. The photo-cured composite, PMPC, which was made from DSMP, presented six times faster curing than pristine epoxy. GCs were found in 97.70–99.63% with a WAC of 2.08–1.95% in the first and fourth week of soaking, respectively. Absence of the T_m_ peak in DSC signposts the stable crystalline nature of sample, with better thermal stability from 7 to 28 days, having 50% wt. loss at 378.2 °C and 440 °C, respectively. As per XRD and FTIR data, HA nucleation in prepared composites indicates the consideration of DS as a suitable material in orthopedics. Amongst all samples, the best results presented by PMPC, either in terms of surface chemistry and filler-matrix interaction or in light of the fact that it promotes HA crystal formation, suggest it as a potential candidate for consideration as bone substitute.

## Data Availability

The original contributions presented in the study are included in the article, further inquiries can be directed to the corresponding author.

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
