# Peer review of "In Vitro Hydroxyapatite Nucleation in Cationically Cured Epoxy Composites with Pulverized Date Seed"

_polymers, 2024, doi:10.3390/polym16243463_

Round 1

Reviewer 1 Report (Previous Reviewer 3)

Comments and Suggestions for Authors

In this work, the authors studied the function of date seed powder as a filler in the preparation of polymer composites for the nucleation of hydroxyapatite crystals to enhance its antibacterial activity. The manuscript is descriptive. Experiments and results are well described. Conclusions follow the results. Therefore, it can be considered for publication, if the following is considered:

-        Correct the references in the text, page 2.

-        Avoid using subtitles with the names of the techniques.

-        Please add a paragraph or a table comparing the achieved results with other methods in the literature.

Author Response

In this work, the authors studied the function of date seed powder as a filler in the preparation of polymer composites for the nucleation of hydroxyapatite crystals to enhance its antibacterial activity. The manuscript is descriptive. Experiments and results are well described. Conclusions follow the results. Therefore, it can be considered for publication, if the following is considered:

Authors are thankful for the appreciation. Point by point response to reviewer's comments is as follows

-        Correct the references in the text, page 2.

Response: Corrected

-        Avoid using subtitles with the names of the techniques.

Response: Corrected

-        Please add a paragraph or a table comparing the achieved results with other methods in the literature.

Response: Added discussion and data (table 1)

Reviewer 2 Report (Previous Reviewer 2)

Comments and Suggestions for Authors

The authors have significantly improved the manuscript and now the role of polymers in obtaining Hydroxyapatite is indicated and explained in detail. As a novelty compared to the previous version of the manuscript, the authors have constructed a Figure (Scheme 1) showing the importance of polymers in their research. The results and discussion are in agreement with each other.

The authors could only enlarge parts a, b and g of Figure 1.

Author Response

The authors have significantly improved the manuscript and now the role of polymers in obtaining Hydroxyapatite is indicated and explained in detail. As a novelty compared to the previous version of the manuscript, the authors have constructed a Figure (Scheme 1) showing the importance of polymers in their research. The results and discussion are in agreement with each other.

Thank you for taking the time to review our manuscript. We appreciate your thoughtful and constructive feedback, which has significantly improved our work.

The authors could only enlarge parts a, b and g of Figure 1.

Response: Modifications have been done, as suggested. 

Reviewer 3 Report (Previous Reviewer 1)

Comments and Suggestions for Authors

This manuscript is a resubmission of a previous manuscript and reports use of pulverized date seed as a filler for incorporation within epoxy polymer composite materials which are then used as substrates for hydroxyapatite formation from a simulated body fluid. The experiments described are clearly relevant to the journal ‘Polymers’ and the results are likely to be of interest to those studying hydroxyapatite formation. Most of the points raised in my review of the earlier version of this manuscript have been adequately addressed. However, there are some points that I feel still need attention and these are detailed below.

1.      Page 2 lines 25-29 “According to existing . . . 20(5) 2035, . . .” These references need to be given in the text using the same in-text citation method as for all the other references and also need to be added to the reference list at the end of the manuscript.

2.      Page 3 line 4 “The purpose of . . . particular objectives.” This sentence is too vague to be meaningful. Either reword it to make it more specific or remove it.

3.      Figure 1. All microscope images ought to have scale bars included with them.

4.      Figure 3 caption. This could be made even clearer by indicating that the images of the unsoaked composites are to the left and the soaked composites to the right.

5.      Page 13 lines 4-5 “. . . indicating higher values of thermal changes in the materials . . .” This statement is very vague and ill-defined. Clarify your actual meaning here. This was left undone from the previous version of the manuscript.

6.       Section 3.3.7. There are multiple instances of distances in millimetres being quoted as flow rates. This discrepancy needs to be corrected. This was left undone from the previous version of the manuscript.

7.      Section 3.3.9. I remain of the view that this section contains significant over-analysis of the data. In Figure 4, most of the bars at each time-point appear statistically indistinguishable based on the error bars given, so I see no justification for the order of antibacterial activity given in the text. Statistical analysis should be used to identify where statistically significant differences exist.

Comments on the Quality of English Language

The manuscript needs some editorial attention to improve the standard of English.

Author Response

This manuscript is a resubmission of a previous manuscript and reports use of pulverized date seed as a filler for incorporation within epoxy polymer composite materials which are then used as substrates for hydroxyapatite formation from a simulated body fluid. The experiments described are clearly relevant to the journal ‘Polymers’ and the results are likely to be of interest to those studying hydroxyapatite formation. Most of the points raised in my review of the earlier version of this manuscript have been adequately addressed. However, there are some points that I feel still need attention and these are detailed below.

Response: The authors express their gratitude for the recognition.

1. Page 2 lines 25-29 “According to existing . . . 20(5) 2035, . . .” These references need to be given in the text using the same in-text citation method as for all the other references and also need to be added to the reference list at the end of the manuscript.

Response 1: Done

2. Page 3 line 4 “The purpose of . . . particular objectives.” This sentence is too vague to be meaningful. Either reword it to make it more specific or remove it.

Response 2: Sentence rephrased

3. Figure 1. All microscope images ought to have scale bars included with them.

Response 3: The document now includes microscopic images in their original format, straight from the instrument.

4. Figure 3 caption. This could be made even clearer by indicating that the images of the unsoaked composites are to the left and the soaked composites to the right.

Response 4: Correction has been made, as suggested

5. Page 13 lines 4-5 “. . . indicating higher values of thermal changes in the materials . . .” This statement is very vague and ill-defined. Clarify your actual meaning here. This was left undone from the previous version of the manuscript.

Response 5: Segment rephrased

6. Section 3.3.7. There are multiple instances of distances in millimetres being quoted as flow rates. This discrepancy needs to be corrected. This was left undone from the previous version of the manuscript.

Response 6: Corrected

7. Section 3.3.9. I remain of the view that this section contains significant over-analysis of the data. In Figure 4, most of the bars at each time-point appear statistically indistinguishable based on the error bars given, so I see no justification for the order of antibacterial activity given in the text. Statistical analysis should be used to identify where statistically significant differences exist.

Response 7: Section has been revised completely and new data set has been added both in graphical and tabulated forms.

Round 2

Reviewer 3 Report (Previous Reviewer 1)

Comments and Suggestions for Authors

The authors have addressed most of the points raised in my earlier review of this manuscript and have added some more data and this has significantly improved the paper. However, there are some still unaddressed points from my earlier review, and I still think that these need to be properly attended to. These points are listed below.

1.      Page 3 line 8 “This research . . . specific objectives.” This sentence is too vague to be meaningful. It should be deleted as it is meaningless filler.

2.     Figure 1. All microscope images ought to have scale bars included with them. Without an idea of the scale of the images, the reader cannot discern how large the features in the image are. Scale bars enable scale to be determined irrespective of the sizing used when viewing the document.

3.  Section 3.2.3. There are multiple instances of distances in millimetres being quoted as flow rates. This discrepancy needs to be corrected. This was left undone from the previous versions of the manuscript.

Author Response

1. Page 3 line 8 “This research . . . specific objectives.” This sentence is too vague to be meaningful. It should be deleted as it is meaningless filler.

Response 1: Sentence has been edited.

2. Figure 1. All microscope images ought to have scale bars included with them. Without an idea of the scale of the images, the reader cannot discern how large the features in the image are. Scale bars enable scale to be determined irrespective of the sizing used when viewing the document.

Response 2: Scale bars have been added.

Section 3.2.3. There are multiple instances of distances in millimetres being quoted as flow rates. This discrepancy needs to be corrected. This was left undone from the previous versions of the manuscript.

Response 3: Word "rate" has been replaced with "value", for easy comprehension.

This manuscript is a resubmission of an earlier submission. The following is a list of the peer review reports and author responses from that submission.

Round 1

Reviewer 1 Report

Comments and Suggestions for Authors

This manuscript reports the use of pulverized date seed as a filler within epoxy polymer composites as a substrate for hydroxyapatite formation from simulated body fluid. The focus of this research is obviously of interest to polymer scientists and is worthwhile in establishing viable biologically-derived materials for hydroxyapatite growth for orthopaedics. The methodology used is mostly reasonable and the results presented show some clear differences between the different materials trialled. However, I feel there are a number of points that the authors need to address, most notably regarding clarity of the presentation of the results, providing a better verification of hydroxyapatite formation and resolving some questionable conclusions drawn from the data. The specific points I feel the authors need to address are detailed below. Assume all points require changes to the manuscript.

1.      All abbreviations used in the manuscript need to be defined at first use. In the abstract, all the abbreviations used need to be defined so that it can be read separately of the main text. The authors’ failure to define abbreviations properly makes the manuscript hard to read and at times quite confusing.

2.      Page 1 lines 23-24 “Gel contents . . . been found supporting.” This sentence is incomplete and so the meaning is quite unclear. Correct this in the manuscript.

3.      Page 1 lines 38-41 “According to existing . . .  development of hydroxyapatite.” This statement needs a supporting reference.

4.      Page 2 lines 46-54 “The purpose of . . . of composites [17, 18].” You need to better describe the intended role of the date seed material, i.e. as a filler in a composite polymer preparation for nucleation of hydroxyapatite. The current text jumps straight to discussing curing methods without connecting it to the earlier discussion on the date seed preparation.

5.      The captions of the figures are inadequate. All figures must be comprehensible independently from the main text. This means that all panels or subsections of figures need to be clearly and succinctly described and labelling of plots and images made clear.

6.      Figure 1 caption. Include mention of the number of replicates used for the measurements in Figure 1d.

7.      Page 3 line 69 “DS were washed . . .” Washed with what? More details are required.

8.      The labelling in Figures 2a and 2b is too small to read easily.

9.      Page 5 line 142 Replace “Upsurge in . . .” with “Intensification of . . .”

10.  Section 3.2.2. I am concerned that you appear to making something of an assumption in concluding that the crystallization observed is hydroxyapatite formation. Whilst this is likely the case, I don’t feel the FTIR and microscopy results alone are sufficient evidence of this. Some unambiguous characterisation confirming hydroxyapatite formation would be advisable.

11.  Page 5 lines 161-162 “The cross-sectional . . . within the matrix.” I don’t think this is really obvious from inspection of the images. It would be useful to point out features in the images that support this assertion.

12.  Figure 3. The labelling within this figure is misaligned and should be corrected.

13.  Page 6 line 179 “Adding filler likely increased heat capacity . . .” This sounds like you are equivocating here. Are you not certain that it has increased the heat capacity?

14.  Section 3.2.4. If the modified powder composites have already degraded at temperatures below 400 °C then how can they demonstrate improved thermal stability above 400 °C?

15.  Page 7 lines 196-197 “. . . indicating higher values of thermal changes in the materials . . .” This statement is very vague and ill-defined. Clarify your actual meaning here.

16.  Page 8 line 223 “. . . 5 wt.% DS . . .” Has this loading of DS powder been optimised?

17.  Page 8 lines 224-225 “The flow rate . . . 23.44 mm . . .” The quantity quoted is a distance not a flow rate.

18.  Page 8 lines 225-226 “. . . all experimental groups . . . in flow rates.” A 4.6-fold drop compared to what?

19.  Page 8 line 228 “. . . the smaller size of the modified samples.” Where are the data indicating that the modified samples had smaller sizes?

20.  Page 8 lines 226-230 “Samples with unmodified . . . filler-matrix interaction.” What are your suggested reasons for the difference seen here between granular fillers and powder fillers?

21.  Section 3.2.8 I think this discussion is really an over-analysis of the data. Based on Figure 1d and the associated error bars it really only appears that the samples with date seed filler are better at suppressing bacterial growth compared to the polymer without filler (PC). All other results appear to be statistically indistinguishable.

Comments on the Quality of English Language

There are typographical errors and problems with syntax that need to be addressed.

Reviewer 2 Report

Comments and Suggestions for Authors

There has been a growing interest in using bio-waste to create valuable materials in recent years due to the focus on environmental sustainability and cost-effectiveness. Researchers are particularly interested in exploring the potential applications of these materials in the biomedical field. One area of current research is the impact of date seed powder on hydroxyapatite formation, specifically focusing on its potential to promote bone health and regeneration. Hydroxyapatite is an essential component of bone tissue and is crucial in maintaining bone strength and structure. This study used date seed in two forms - grains and powder - with unmodified and modified surface chemistries. The prepared composites were tested in vitro by soaking them in simulated body fluid. The results from FTIR confirmed the formation of hydroxyapatite in all soaked samples. Additionally, TGA results indicated improved thermal stability after soaking, suggesting a higher concentration of hydroxyapatite.

The strength of the manuscript is reflected in a large number of experimental methods that provide mutually agreeable results.

The weakness of the manuscript is that the importance of polymers or polymerization during the test is not emphasized, the goals, especially the role of polymers, are not clearly emphasized, the discussion should be more coherent.

Suggested corrections:

1. In the introductory part, expand and emphasize how the polymerization process fits into the author's research.

2. Explain all abbreviations.

3. In the discussion, give the scheme of the formation of hydroxyapatite, where the influence of the date seed powder is also included, as well as the polymerization process, which needs to be emphasized

4. List the relative errors in Table 1.

Reviewer 3 Report

Comments and Suggestions for Authors

In this work, the authors prepared and characterized Hydroxyapatite from date seed. The manuscript  is reasonably clear, the experiments and the results are more or less described, and the conclusions follow the results, therefore, it can be considered for publication if the following major revision is considered:

-        Please indicate the meaning of HA, DS, FTIR, and TGA in the abstract.

-        Please explain the reason for presenting a section of materials when this article is a review. Is Figure 1 original results or is this a citation of a published article? If yes, please include citations and ask for copyright permissions.

-        Definitively, this article is not a review! Please change the type of article at the beginning of the article.

-        In section 2.4, please indicate the meaning of FTIR-ATR.

-        In line 105, the authors wrote “Ws = initial weight of sample, Wd = difference of sample weight before and after gel content.” Please create a clear sentence with this information. For example , where Ws  is the  initial weight of sample, Wd is the difference of sample weight before and after gel  content. The same for the information in line 112, Wt = weight of composite after water absorbance, Wo = weight of original composite.

-        Please clarify the meaning of Water Absorbance in the manuscript. Don't the authors mean adsorbed water or water content? Please clarify better how the water absorbance in percentage was obtained because weight is not absorbance. Absorbance is the quantity of light absorbed by the medium and the adsorption of water molecules leads to sample weight increasing.

-        Please include error bars in the values listed in Table 1.

-        In the section Gel Contents & Water Absorbance Capacity, please indicate the meaning of Water Absorbance Capacity. Don't the authors mean water adsorption capacity?

-        Use abbreviations in Table 1 and indicate the meaning of these abbreviations in the caption of Table 1 or at the end of this Table. For example, Maximum can be changed to Max, Gel content by GC, Water absorbance by WA,…

-        Why the antibacterial activity was presented in Figure 1? Please describe better the achieved results and show this figure in this section.